# Age- and Gender-Related Differences in the Morphology of Cuff Tear Arthropathy: A Cross Sectional Analysis

**DOI:** 10.3390/jfmk8010008

**Published:** 2023-01-11

**Authors:** Michael Stephan Gruber, Martin Bischofreiter, Patrick Brandstätter, Josef Hochreiter, Patrick Sadoghi, Reinhold Ortmaier

**Affiliations:** 1Department of Orthopedic Surgery, Ordensklinikum Linz Barmherzige Schwestern, Vinzenzgruppe Center of Orthopedic Excellence, Teaching Hospital of the Paracelsus Medical University, 4020 Linz, Austria; 2Trauma Center Linz, Teaching Hospital of the Paracelsus Medical University Salzburg, 4020 Linz, Austria; 3Department of Orthopaedics and Trauma, Medizinische Universität Graz, 8010 Graz, Austria

**Keywords:** osteoarthritis, shoulder, glenohumeral joint, rotator cuff tear arthropathy, shoulder anatomy, morphology of cuff tear arthritis

## Abstract

Rotator cuff tear arthropathy (CTA) is the most common reason for reverse total shoulder arthroplasty (RSA). There is minimal understanding of the natural progression of osteoarthritis of the shoulder and of the morphologic differences between men and women and between younger and older patients. This trial comprised 309 patients (342 shoulders) who underwent RSA due to CTA in the period between January 2009 and September 2019. The patients were divided into gender and age groups. Preoperative X-rays, computed tomography and magnetic resonance imaging were conducted using various classifications to describe the morphology of the CTA. Of all 342 analyzed shoulders, 209 were right and 133 were left shoulders. A total of 257 female shoulders and 85 male shoulders were assessed. Both mean age and age distribution were significantly different (74.37 years in female and 70.11 years in male patients, *p* = 0.001; 70.2% female patients in the age group <75.5 years and 80.1% in the age group >75.5 years, *p* = 0.045). A larger extent of progression of the fatty infiltration was detected both in the female cohort (*p* = 0.006) and in the older age group (*p* = 0.001). Additionally, older patients had significantly higher levels of muscle retraction (Patte; *p* = 0.003), a lower acromiohumeral distance (*p* = 0.042) and more advanced CTA (Seebauer; *p* = 0.006).

## 1. Introduction

Cuff tear arthropathy (CTA) is one of the secondary forms of shoulder arthritis and the main indication for reverse shoulder arthroplasty (RSA). Despite the extensive knowledge of this kind of pathology and its prevalence according to age and sex, there is still little information about the natural progression and development of CTA. Furthermore, no studies exist on the differences in the morphologies of CTA between females and males as well as younger and older patients. Therefore, we analyzed the age- and gender-related morphological differences of CTA in order to gain deeper insight into CTA progression.

In the nineteenth century, the pathology was commonly known as chronic rheumatic arthritis, until in 1983, Neer et al. first introduced the term “arthropathy of the rotator cuff” [1]. Several theories explained the development of CTA and its close relation to rotator cuff tear (RCT) [2,3,4,5,6]. Rotator cuff deficiency results in the humeral head moving position into an anteroinferior subluxation and therefore leads to arthritic changes, as Walch et al. have shown [7]. In 2020, Van Parys et al. found a difference of the coracoacromial complex between CTA, glenohumeral osteoarthritis and non-pathological shoulders [8]. However, despite the well-known pathogenesis of CTA and the underlying mechanisms of a rotator cuff tear, the question of why patients develop different glenoid defects remains unclear. To date, there has been no scientific examination of the reasons for these differences. Thus, the purpose of this study was to investigate possible age- and gender-related irregularities of the bony structures and soft tissue of the glenohumeral joint, which would lead to an increased understanding of the background of CTA.

The purpose of this study was to radiologically investigate patients suffering from CTA in terms of morphology depending on age or gender.

## 2. Materials and Methods

This retrospective cohort study was conducted to investigate age- and gender-related differences in CTA morphology. Patients who received RTSA for CTA in the period between January 2009 and September 2019 with preoperative computed tomography and magnetic resonance imaging were included. The exclusion criteria were: (1) fractures affecting the glenohumeral joint; (2) pre-existing rheumatic disorders; and (3) revision surgery. A total of 309 patients, adding up to 342 shoulder treatments, from one single institution, were included after screening for eligibility. Approval from the Institutional Review Board was obtained prior to the investigation (1018/2020). Informed consent from all subjects involved in the study was provided.

Preoperative X-rays, computed tomography and magnetic resonance images of every patient were assessed using common classifications. These were from the works of: (1) Seebauer [9] and (2) Hamada [10] (classifications for rotator cuff tear arthropathy); (3) Favard [11] and (4) Walch [12] (classifications for glenoid configuration); and (5) Habermeyer [13], (6) Goutallier [14], (7) Thomazeau [15] and (8) Patte [16] (classifications for muscular conditions). Additionally, (9) the acromiohumeral index (AHI) [17], (10) the acromiohumeral distance (AHD), (11) the critical shoulder angle (CSA) [18] and (12) the version of the glenoid were measured to evaluate morphologic differences between both groups.

Patients were then assigned into groups according to gender, resulting in one female (*n* = 257) and one male group (*n* = 85). For the assessment of age-related differences, patients were divided into younger and older age groups. The divisor was set as the median age of the trial population at the time of surgery (74.5 years). The twelve different classifications and measurements were then compared to determine differences.

Data were analyzed using IBM SPSS Statistics (Windows, 64-bit, version 23.0; IBM Corp., Armonk, NY, USA). Metrical variables were tested using the Kolmogorov–Smirnov test to check for normal distribution. A two-way ANOVA and Mann–Whitney *U* test were then performed to analyze metrical dependent variables. To evaluate the nominal variables, a Chi-square test was performed. The statistical significance was set at *p* < 0.05. The effect size was calculated for significant differences (r < 0.3 = low effect size; 0.3 ≤ r ≤ 0.5 = medium effect size; r > 0.5 = large effect size).

## 3. Results

In total, 342 shoulders were examined in this study, of which 133 were left and 209 were right shoulders. There were 257 female and 85 male shoulders. The demographic data for each group are shown in Table 1. A significant difference was found between both gender groups—female patients were significantly older than male patients (r = 0.18)—and between both age groups—the proportion of female patients was higher in the older age group (r = 0.12). While X-rays were available for 300 shoulders, MRI was performed preoperatively on 166 shoulders and a CT scan on 77 shoulders.

Our analysis of the patients’ data is shown in Table 2. When comparing female with male patients, fatty degeneration according to Goutallier showed significantly lower grading in the males and higher grading in the female population (*p* = 0.006; r = 0.29). The other data did not show any significant difference; however, men tended to have greater AHD and the glenoid version, although both were not significant.

On the other hand, the analysis according to the age groups showed several significant differences. Firstly, a greater AHD was measured in younger patients’ shoulders (*p* = 0.001; r = 0.11; Figure 1). Secondly, CTA classified according to Seebauer [9] showed a larger extent of progression among the older population, with more patients graded at 2A and 2B (*p* = 0.006; r = 0.21). Thirdly, greater damage of the rotator cuff combined with more retraction and higher fatty infiltration in older compared to younger patients was detected—all four classifications of Habermeyer [13] (*p* < 0.001; r = 0.31), Goutallier [14] (*p* = 0.001; r = 0.33), Thomazeau [15] (*p* = 0.037; r = 0.21) and Patte [16] (*p* = 0.003; r = 0.29) showed significantly greater damage of the rotator cuff in older patients.

## 4. Discussion

In this study, we evaluated 309 patients (342 shoulders) with CTA according to popular classifications. The aim of this study was to investigate possible age- and gender-related irregularities of the bony structures or the soft tissue of the glenohumeral joint to better understand CTA.

As expected, our data confirmed the natural progression of cuff tear arthropathy over time. The examination considering age showed significant differences of AHD, the progression of the CTA, according to the classification of Seebauer, and greater damage of the rotator cuff combined with more retraction and higher fatty infiltration in older patients. The younger patients had a significantly greater AHD, lower stages in the Seebauer classification (more patients graded 2A and 2B) and less soft tissue damage than the older population according to Habermeyer, Goutallier, Thomazeau and Patte. These results support the widely known principle of the natural degeneration of the cuff tear in older people and its progression over time.

Furthermore, we found some interesting differences between the female and male populations. The examined female shoulders showed significantly more fatty degeneration of the rotator cuff muscles than the male ones according to Goutallier’s classification (*p* = 0.006). We conducted a power calculation according to Hoenig and Heisey [19], which showed a power of >80%.

We suggest that this is related to a higher mean age of the female patients and hence greater progression of fatty degeneration. A study conducted in 2010 by Oh et al. concluded that “older age is correlated with higher fatty degeneration grade”, supporting this theory [20]. As described, the patient population was significantly different regarding age at surgery; male patients were on average 70.11 years old, whereas the female patients were on average 74.37 years old. No other gender-related differences were found, except for fatty degeneration, and female and male patients showed the same progression of CTA when we saw them for the first time. However, despite the younger mean age, the AHD tends to be lower in male patients. Gelvosa et al. suggested gender-related differences with regard to AHD, but stated that rather female patients had a lower acromiohumeral distance [21].

Our trial included 257 female and 85 male shoulders. This appears to be consistent with the existing literature, which states that the prevalence of CTA is higher among women than men [22,23]. We can only assume that this result is of multifactorial genesis and highly dependent on local conditions, including retirement age, life expectancy and the distribution of the population, since the prevalence of RCT shows the opposite trend, with men making up the majority of the patient population [24,25].

The data suggest that female and male patients both suffer from progressing CTA to the same extent, but female patients receive surgery at an overall higher age due to their superior health conditions compared to male patients in the same age groups. However, this is not statistically significant; hence, this only represents the personal opinion of the authors.

Our study had several strengths. With 342 shoulders, a large number of patients were evaluated in this study. Furthermore, data over a period of ten years were analyzed. Internationally respected and commonly used classification methods were used to make a good comparison between our results and the results of existing and future studies.

Nevertheless, our study did have some limitations too. The trial was conducted retrospectively, and only radiographic data were analyzed. Neither pre- nor postoperative clinical scores were included. Due to its retrospective study design, many patients only had either X-rays and MRI or X-rays and a CT scan. Finally, our study lacked a control group.

## 5. Conclusions

Summarizing the findings, female patients account for the majority of RSA cases due to CTA. Female patients undergo the surgery at a higher age. CTA seems to be progressive, which is underlined by our findings that there were significantly higher gradings in six out of 12 classifications in the older population.

## Figures and Tables

**Figure 1 jfmk-08-00008-f001:**
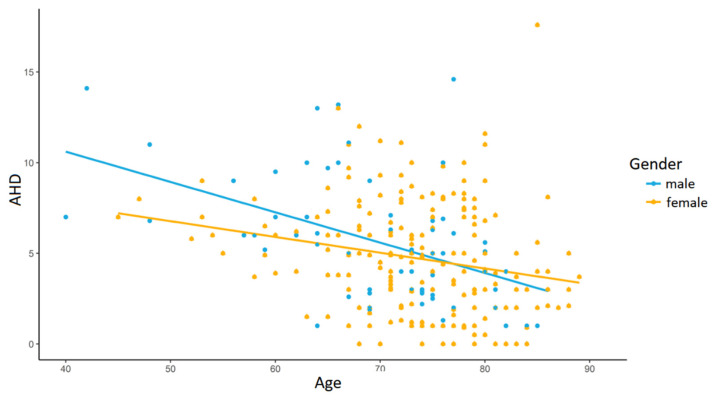
Gender-related differences in acromiohumeral distance. The linear regression model shows a relation between acromiohumeral distance and progressing age (*p* = 0.001; r = 0.11).

**Table 1 jfmk-08-00008-t001:** Demographic data of the patient population ^a^.

*n* = 342	Female (*n* = 257)	Male (*n* = 85)	*p*
Age (years)	74.37 ± 7.5	70.11 ± 10.0	0.001
Side (left/right)	101 (39.3)/156 (60.7)	32 (37.6)/53 (62.4)	0.887
***n* = 342**	**<74.5 years (*n* = 171)**	**>74.5 years (*n* = 171)**	** *p* **
Gender (female/male)	120 (70.2)/51 (29.8)	137 (80.1)/34 (19.9)	0.045
Side (left/right)	65 (38)/106 (62)	68 (39.8)/103 (60.2)	0.824

^a^ Data are presented as the mean ± standard deviation or absolute numbers (percentages).

**Table 2 jfmk-08-00008-t002:** Interpretation of the statistical analysis of nominal and metrical variables ^a^.

	Distribution							Statistical Significance ^f^	
	*n*	Female	Male	Older	Younger	Median	SD ^e^	f/m	o/y
Seebauer (1A/1B/2A/2B)	341	256	85	171	170			0.341	0.006
Hamada (1–5)	329	248	81	166	163			0.483	0.417
Favard (E0–E4)	342	257	85	171	171			0.807	0.534
Walch (A1–2/B1–B3/C/D)	238	180	58	106	132			0.686	0.107
Habermeyer (0/A/B/C)	159	117	42	61	98			0.652	<0.001
Goutallier (0–4)	166	121	45	63	103			0.006	0.001
Thomazeau (1–3)	183	121	62	62	103			0.063	0.037
Patte (0–3)	163	119	44	62	101			0.543	0.003
AHI ^b^	287	214	73	144	143	0.71	0.11	0.219	0.397
AHD ^c^ (mm)	272	203	69	126	146	4.9	3.2	0.076	0.001
CSA ^d^ (°)	294	220	74	150	144	31.76	4.95	0.843	0.123
Version (°)	232	178	54	103	129	5.6	8.8	0.144	0.482

^a^ Numbers are presented as total count, with grades or units in brackets. ^b^ Acromiohumeral Index. ^c^ Acromiohumeral distance. ^d^ Critical shoulder angle. ^e^ SD, standard deviation. ^f^ f, female; m, male; o, older population; y, younger population.

## Data Availability

The data and materials this study is based on are available from the corresponding author, R.O.

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
