# Peer review of "Age- and Gender-Related Differences in the Morphology of Cuff Tear Arthropathy: A Cross Sectional Analysis"

_jfmk, 2023, doi:10.3390/jfmk8010008_

Round 1
Reviewer 1 Report
The authors submitted a manuscript that was a cross-sectional analysis of age and gender related differences in the morphology of cuff tear arthropathy.
Rotator cuff tear arthropathy represents a spectrum of shoulder pathology characterized by rotator cuff insufficiency, diminished acromiohumeral distance with impingement syndromes, and arthritic changes of the glenohumeral joint. Massive rotator cuff tears may lead to the development of cuff tear arthropathy. Although this pathology has been recognized for more than 150 years, treatment strategies have continued to evolve. The authors summarized a decade of data. Overall, this manuscript is acceptable.
Nonetheless, there are some issues that need to be noted.
1. The manuscript lacks references published in recent years.
2. There are some errors in grammar and format in the whole manuscript: inconsistencies; tense; single and plural expressions; the use of prepositions and definite/indefinite articles; uppercase and lowercase; abbreviations; punctuation.
Author Response
The authors submitted a manuscript that was a cross-sectional analysis of age and gender related differences in the morphology of cuff tear arthropathy.
Rotator cuff tear arthropathy represents a spectrum of shoulder pathology characterized by rotator cuff insufficiency, diminished acromiohumeral distance with impingement syndromes, and arthritic changes of the glenohumeral joint. Massive rotator cuff tears may lead to the development of cuff tear arthropathy. Although this pathology has been recognized for more than 150 years, treatment strategies have continued to evolve. The authors summarized a decade of data. Overall, this manuscript is acceptable.
Nonetheless, there are some issues that need to be noted.
- The manuscript lacks references published in recent years.
A thorough review of the current literature was done and recent data was added (References numbers 1, 7 & 8)
- There are some errors in grammar and format in the whole manuscript: inconsistencies; tense; single and plural expressions; the use of prepositions and definite/indefinite articles; uppercase and lowercase; abbreviations; punctuation.
Paid english language editing was done using the MDPI editing service.
Reviewer 2 Report
The authors presented a cross-sectional analysis comprising 309 patients who underwent rotator cuff tear arthropathy due to reverse total shoulder arthroplasty. They concluded interesting logical findings, including but not limited to: the progression of the fatty infiltration higher in females and older age groups. Interestingly, they found older patients had significantly more muscle retraction, lower acromiohumeral distance, and more advanced CTA, which would be expected to occur during the aging process.
Author Response
The authors presented a cross-sectional analysis comprising 309 patients who underwent rotator cuff tear arthropathy due to reverse total shoulder arthroplasty. They concluded interesting logical findings, including but not limited to: the progression of the fatty infiltration higher in females and older age groups. Interestingly, they found older patients had significantly more muscle retraction, lower acromiohumeral distance, and more advanced CTA, which would be expected to occur during the aging process.
Thank you for your review and opinion.